# Motion Control Method of Bionic Robot Dog Based on Vision and Navigation Information

**Zhaolu Li [1], Ning Xu [2], Xiaoli Zhang [1,\*], Xiafu Peng [1] and Yumin Song [3]**

[1] School of Aerospace Engineering, Xiamen University, Xiamen 361102, China; a1148963175@sina.com (Z.L.); xfpengxmu@126.com (X.P.)

[2] Shandong Academy of Agricultural Machinery Sciences, Jinan 252100, China; xuning7608@163.com

[3] School of Automotive Engineering, Shan Dong Jiaotong University, Jinan 250357, China; songyumin@sdjtu.edu.cn

\* Correspondence: zhangxl_xmu@163.com

**Abstract:** With the progress and development of AI technology and industrial automation technology, AI robot dogs are widely used in engineering practice to replace human beings in high-precision and tedious industrial operations. Bionic robots easily produce control errors due to the influence of spatial disturbance factors in the process of pose determination. It is necessary to calibrate robots accurately to improve the positioning control accuracy of bionic robots. Therefore, a robust control algorithm for bionic robots based on binocular vision navigation is proposed. An optical CCD binocular vision dynamic tracking system is used to measure the end position and pose parameters of a bionic robot, and the kinematics model of the controlled object is established. Taking the degree of freedom parameter of the robot's rotating joint as the control constraint parameter, a hierarchical subdimensional space motion planning model of the robot is established. The binocular vision tracking method is used to realize the adaptive correction of the position and posture of the bionic robot and achieve robust control. The simulation results show that the fitting error of the robot's end position and pose parameters is low, and the dynamic tracking performance is good when the method is used for the position positioning of control of the bionic robot.

**Keywords:** visual navigation; bionic robot dog; robustness; control

## 1. Introduction

In the process of humans' subjective initiative, there are many dangerous situations that human beings cannot safely perform, such as forest fire inspection, planet detection, military material transportation, earthquake relief, nuclear field experiments with pollution, mine clearance and explosion removal, and deep-sea field exploration. Biomimes play an irreplaceable role in performing this series of difficult and dangerous tasks [1]. However, the relevant technology for bionic mechanical dogs is currently not mature, and all countries are continuously increasing their research and development investments in this field. Therefore, the research on bionic mechanical dogs has great practical significance for improving national soft and hard strength and human security, and it has important practical application value.

Bionic robot technology is a product that covers many disciplines. Robots can currently be divided into wheeled robots, crawler robots, foot robots and so on according to the difference in the actuator [2]. The bionic robot dog studied in this paper belongs to the foot-type robots. Because the landing point of the bionic robot dog is discrete and controllable, this feature makes the bionic robot dog able to cope with different geographical environments. In addition, the unique bionic leg structure of the bionic robot dog greatly increases the degree of freedom of the whole system, so the bionic robot dog is more stable and flexible in complex road conditions [3]. The bionic robot dog has a high requirement for the accuracy of pose positioning and tracking and needs to reach a millimeter level of

error to achieve accurate control [4]. Therefore, it is necessary to optimize the control of the robot dog, reduce the geometric error of the robot dog, and improve the positioning and tracking ability of the robot dog's end posture [5]. Research on robust control of robot dogs is of great significance to improving the operating accuracy and efficiency of robot dogs [6].

The bionic robot dog is affected by spatial disturbance factors in the process of determining the pose, and the kinematic model of the robot dog is a multicoupling nonlinear model, which is difficult to control effectively and is prone to control errors [7]. Accurate visual navigation and calibration solve the kinematic equations of the robot dog, realize the optimal solution of the controlled object parameters of the robot dog, and realize robust control. The traditional robust control methods for bionic robot dogs mainly include the fuzzy PID control method, pose fusion filter control method, synovial integral control method, inversion control method, etc., so that the controller parameters are adaptively adjusted in a limited number of linear models, and combined with kinematic loop weighting, the robot dog's end pose positioning is achieved, and a certain control efficiency is achieved [8]. Some scholars have proposed a bionic robot dog control algorithm based on a variable structure PID fuzzy neural network. The Smith structure is used to design the object structure model of bionic robot dog control, and the forward three-layer adaptive PID neural network model is used as a learner to realize the bionic robot dog [9]. The control optimization improves the position and pose control accuracy of the robot dog, but the calculation cost of the algorithm is high, and the real-time performance of the robot dog position and pose adjustment is poor [10].

Some scholars have used the improved closed-loop detection algorithm based on the spatial position uncertainty constraint to control the motion state of the robot dog [11]. The outer control loop is designed with the idea of synchronous phase modulation control, and the closed-loop detection model is designed with a DC/AC inverter model. The tracking and identification of the dog path can improve the robustness of the bionic robot dog grasping control, but this method is vulnerable to small disturbance factors, resulting in output measurement error [12]. Some scholars have used the improved artificial potential field method for obstacle avoidance and the path planning design of the robot dog, and they have proposed an improved artificial potential field method with relative speed for fuzzy adaptive command filtering, ensuring that the robot dog can escape the minimum trap and improving the path tracking accuracy of the bionic robot dog, but this method is prone to pose tracking error when there is disturbance in the virtual auxiliary arm [13]. Some scholars have realized the torque balance control of the robot dog in pitch, roll, and yaw directions using the bionic robot dog running gait optimization control method based on energy efficiency optimization, but this method can easily cause the posture parameters to fall into the local optimal solution [14]. Some scholars have used the ant colony algorithm with dynamic search strategy to control the path planning of the robot dog and dynamically adjusted the threshold of the ant colony algorithm to expedite the convergence speed, but the planning accuracy is not high.

To solve these problems, this paper proposes a robust control algorithm for bionic robots based on binocular vision navigation. First, the binocular vision dynamic tracking system is used to collect the pose parameters and build the kinematics model; then, we use the binocular vision tracking method to realize the adaptive correction of the position and posture of the bionic robot and the improved design of the control law; and finally, simulation experiments are performed to demonstrate the superior performance of this method in improving the navigation and tracking accuracy of the robot and reducing the end position and pose control error.

## 2. Materials and Methods

### 2.1. Kinematics Analysis

The kinematic model of the bionic mechanical dog is mainly aimed at the following problems:

(1) When the relevant parameters of each mechanism of the mechanical dog's leg joints have been obtained, the coordinates of the end point of the mechanical dog's joint in

the entire body coordinate system can be calculated according to the joint rotation angle, that is, the position and posture, which is a positive motion problem; and

(2) When the parameters related to each mechanism of the mechanical dog's leg joints have been obtained, we can analyze the rotation angles of each joint through inverse kinematics according to the information at the end of the leg joints, which is the inverse kinematics problem.

The common gaits of bionic robots mainly include crawl, trot, pace and bound. Crawl is a common gait of some reptiles. A robot in a crawling state will always have at least three legs or four legs, as shown in Figure 1. One leg is lifted, and the other three legs form a stable triangle area (FL, FR, HL) for the support leg, so the crawling gait has high stability and strong robustness. FL, FR, HL and HR represent the left foot, right foot, left hand, and right hand respectively.

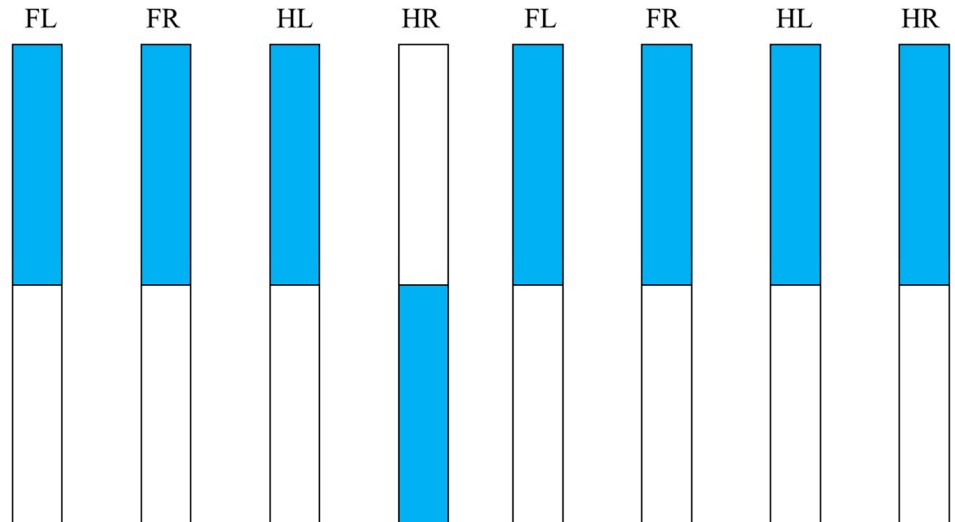

**Figure 1.** Schematic diagram of crawl.

The motion planning problem of the bionic robot has always been a core and difficult point of robot motion. Its essence is to continuously solve and optimize a continuous path from the initial position and pose point to the final destination position and pose point according to the target movement position and geographical environment of the robot [15]. In this paper, the bionic mechanical dog is abstracted as a position and pose point, and its essence is the position and pose of the mechanical dog in the Cartesian coordinate system. Further, we believe that each pose point corresponds to the mechanical dog. One by one with the actual leg joint structure, through the analysis of each gait in the previous section, to greatly improve the walking ability of the bionic mechanical dog in a complex geographical environment, this paper uses the crawling gait as the main gait of movement.

In this crawling sequence, the quadruped support phase is not considered, and the quadruped grounding phase is formed instantly after the swing phase lands. At this time, the stability of the system is greatly enhanced, but at the same time, the system reduces the running speed of the system. When the swing phase of the robot dog switches from the front left leg to the rear right leg, the center of gravity of the robot dog switches from one side of the support field to the other side. During this process, the center of gravity of the robot dog is closest to the edge of the support field. Therefore, the stability of the mechanical dog is very poor at this time. At this time, we hope to add a quadruped support phase. Similarly, when the front right leg is switched to the rear left leg, the center of gravity of the triangular support domain also switches from one side to the other side, and it is also necessary to add quadruped support.

COG is the center of gravity of the bionic mechanical dog. The early static stability determination method is used to cause the projection of the COG on the ground not to

exceed the closed polygon formed by the contact between the mechanical dog and the ground. Before using this criterion, it is necessary to ensure that there are enough closed polygons. This criterion was applied in robots, such as Honda's P1, in the early days and achieved good results. For a quadruped robot, such as a bionic mechanical dog, we can use the static balance stability margin, that is, static stability margin, abbreviated SSM. When the bionic mechanical dog is walking, it must ensure the distance between the COG and the closed area boundary formed by the supporting feet. The minimum distance is greater than 0, so at this time, the bionic mechanical dog must continuously adjust the COG to meet the static stability condition. This continuous judgment of the COG position during the walking process will cause the robotic dog to walk slowly. Therefore, when the robot has high speed requirements, it cannot meet the requirements.

Therefore, the control method of zero-point torque is introduced; that is, ZMP (zero moment point) is the first theory proposed for the study of the stability of the footed robot. This theory has also become the basis for many researchers to study the stability of biped robots, such as QRIO developed by Sony, ASMIO developed by Honda, and WABOT developed by Waseda University in Japan. With the continuous development of quadruped robot research, ZMP began to develop. It has gradually occupied an important position in the field of stability determination of quadruped robots.

ZMP is the point at the intersection of the resultant force of various external forces on the bionic mechanical dog and the closed area formed by the intersection of the bionic mechanical dog and the ground. ZMP can be considered as the gravity and the intersection point of inertial force; we can regard the shortest distance between the zero-moment point and the boundary of the above closed area as the dynamic balance stability margin SZMP, so we hope that the dynamic balance stability margin value always satisfies the condition SZMP > 0 to ensure that, when moving, the bionic mechanical dog can walk stably without rolling and other phenomena. When SZMP > 0 is not satisfied, it can be considered that the intersection point of the bionic mechanical dog and the inertial force have exceeded the boundary of the closed area. We believe at this time that the ZMP is inappropriate; that is, the bionic mechanical dog is in an unbalanced state.

After the above kinematic analysis and dynamic analysis, the following block diagram of the position control principle can be obtained. According to the end position planned by the trajectory, the target position of each joint motor can be obtained through PID control, and then each joint can calculate the current position through vector control. The required current id and iq drive the joint motion, and the principal structure of position control is shown in Figure 2.

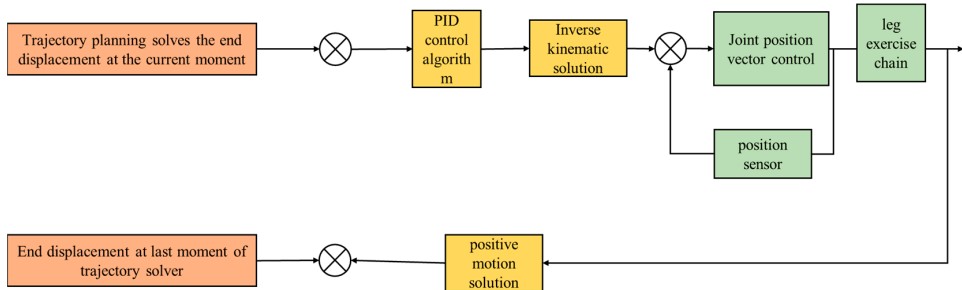

**Figure 2.** Block diagram of the principal structure of position control.

## 2.2. ANSYS Finite Element Analysis

ANSYS is finite element analysis software. If the model in Solidworks is directly imported into ANSYS, the details of the 3D model of the bionic mechanical dog will complicate the meshing, so in the actual analysis, we need to eliminate the features that have little effect on the finite element performance. The purpose of static analysis is to determine the stress distribution and mechanical structure deformation of the bionic mechanical dog under the action of its own gravity so as to facilitate engineering and technical personnel

in checking whether the stiffness and strength of the bionic mechanical dog structure can meet the level of our design according to the results of the finite element analysis.

The quality of pretreatment in finite element analysis determines the accuracy of the final finite element analysis results. Table 1 shows the material and mechanical properties of the bionic mechanical dog structure.

**Table 1.** Material mechanical parameter.

| Material | Density (kg cm$^3$) | Elastic Coefficient/GPa | Fatigue Strength/MPa | Poisson's Ratio |
|---|---|---|---|---|
| Aluminum alloy 6060 | 2.7 | 68.9 | 62.1 | 0.33 |
| 45 # Steel | 7.85 | 210 | 127 | 0.26 |

The 3D model of the leg joint established by Solidworks is shown in Figure 3.

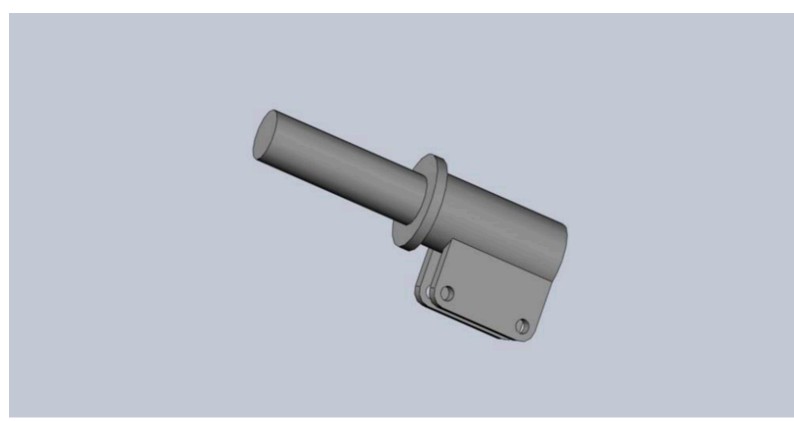

**Figure 3.** Leg joints model in Solidworks.

The force of the lower leg joint is complex, and the moment it bears under various working conditions is large. Its fatigue characteristics and the size of the force on the stability of the whole system are very important. Therefore, we need to perform static analysis on the lower leg joint to analyze the corresponding strain and stress. The ANSYS static analysis of the lower leg joint is shown in Figure 4. The results of ANSYS simulation analysis show that the deformation and stress of the lower leg meet our requirements.

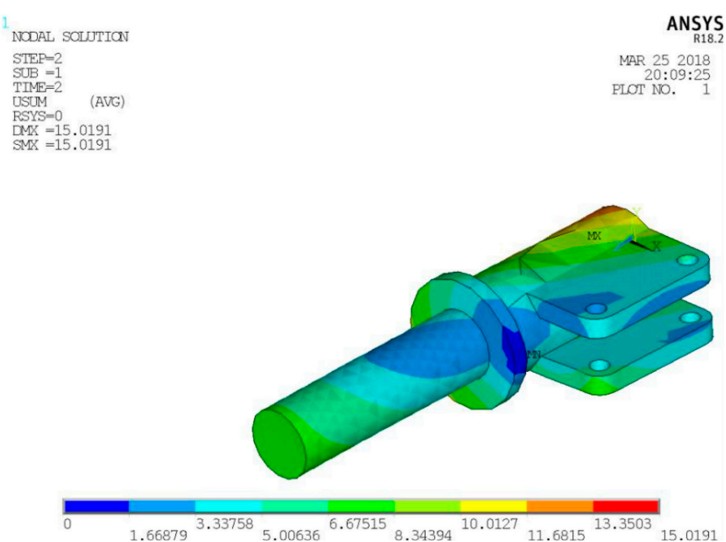

**Figure 4.** Stress analysis of leg joint.

After loading the meshed leg model, the stress distribution is shown in Figure 4. The maximum stress is 15.0191 MPa, and the maximum stress point is at the lower leg cylinder wall, as shown in the red area in Figure 4. However, 15.0191 MPa meet the yield strength limit of aluminum material, so it can be considered that the design meets the stress and strain requirements.

## 3. Results and Discussion

### 3.1. Kinematics Model Analysis of the Controlled Object

To realize the robust tracking control of the binocular vision navigation of the bionic robot, it is necessary to first build the kinematics model of the robot and build the controlled object and control constraint parameter model. The optical CCD binocular vision dynamic tracking system is used to measure the end position and pose parameters of the bionic robot. Figure 5 shows the geometric structure of the bionic robot dog.

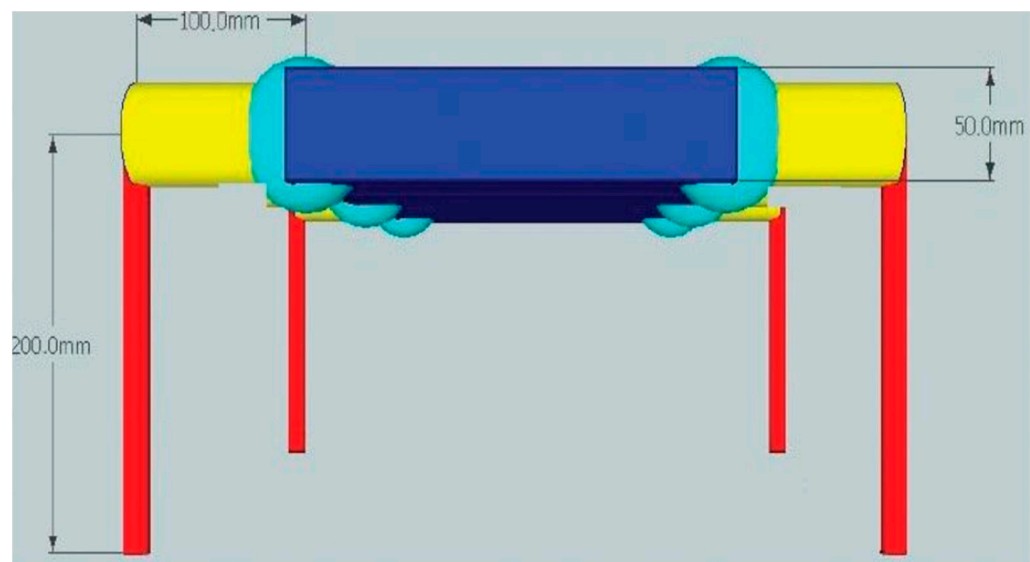

**Figure 5.** Front view of biomimetic mechanical dog.

The optical CCD binocular vision dynamic tracking system is used to measure the pose parameters and data collection of the robot, the pose parameters of the robot are collected in the six degrees of freedom of the bionic robot, and the determined model set $M$ = $\{m_i \mid i = 1, 2,..., m\}$ represents the work space motion attribute model of the bionic robot, such as walking, golloping, pitching, rolling, etc. The discrete form of the robot motion state equation and the observation equation of the pose parameters are obtained:

$$\begin{cases} x(k+1) = \Phi_i(k)x(k) + w_i(k), i = 1, 2, \cdots, m \\ z(k) = H_i(k)x(k) + v_i(k), i = 1, 2, \cdots, m \end{cases} \quad (1)$$

Among these values, $w_i(k)$ and $v_i(k)$ are the poses of the joints and the disturbance noise that affects the robot's pose determination, respectively. Inverse kinematics are used to perform the dynamic tracking of the robot's binocular vision, and the resulting disparity measure information consists of $Q_i(k)$ and $R_i(k)$. In the known $m$ pose dynamic distribution model, the initial state of the robot's rotational joint is $x^i(0) = \hat{x}^i(0)$ and the forward kinematics correlation characteristic of the link structure is $u_i(0) = P(m_i(0)/z(0))$. The kinematic loop structure model of the robot is constructed in three-dimensional space, and the motion state feature transition probability matrix is obtained as $[P_{ij}]$, with $P_{ij}$ representing the transfer probability from pose state $m_i$ to $m_j$ in the kinematic loop model. From time $k - 1$ to time $k$, bindinate system to obtain the measurement output of position and pose parameters:

$$Z_c \begin{bmatrix} U \\ V \\ 0 \end{bmatrix} = \begin{bmatrix} \frac{f}{d_x} & 0 & U_0 & 0 \\ 0 & \frac{f}{d_y} & V_0 & 0 \\ 0 & 0 & 1 & 0 \end{bmatrix} \begin{bmatrix} R & T \\ O^{\mathrm{T}} & 1 \end{bmatrix} \begin{bmatrix} X_w \\ Y_w \\ Z_w \\ 1 \end{bmatrix} \tag{2}$$

Among these values, $Z_c$ represents the vertical distance from the left and right cameras to the end effector, $(x_w, y_w, z_w)$ represents the pose relationship between the links, $(U_0, V_0)$ represents the pixel coordinates of the center of the binocular vision measurement image, and $(U, V)$ represent the point coordinates of joint independent motion.

On the basis of collecting the pose parameters of the robot, the kinematic model of the controlled object is established, and the 6-DOF parameters of the rotating joints of the robot are used as the control constraints. Based on the extended Kalman filtering method, the link transition formula of the bionic robot is obtained:

$$^{i-1}T_i = \begin{bmatrix} c\theta_i & -s\theta_i & 0 & a_{i-1} \\ s\theta_i c\alpha_{i-1} & c\theta_i c\alpha_{i-1} & -s\alpha_{i-1} & -d_i s\alpha_{i-1} \\ s\theta_i s\alpha_{i-1} & c\theta_i s\alpha_{i-1} & c\alpha_{i-1} & d_i c\alpha_{i-1} \\ 0 & 0 & 0 & 1 \end{bmatrix} \tag{3}$$

Among these values, $s$ represents the sine of angle $\theta$, and $c$ represents the cosine of angle $\theta$. The angle of the binocular visual navigation and tracking gyroscope is used as the predicted angle, and the measurement deviation is $\omega(k)$. In the 6-DOF space model, the transition of each link is multiplied to obtain the pose transition matrix of the bionic robot:

$$^0_6T = ^0_1T ^1_2T ^2_3T ^3_4T ^4_5T ^5_6T \tag{4}$$

Equation (4) represents the pose matrix of the robot end effector. The dynamic variable of the selected pose angle is $x = [\varphi, \varphi, \theta]^T$, and the dynamic equation when the bionic robot is in a nonlinear motion state is expressed as $x = f(x, u)$, under the standard pose structure. The state quantity is $x_0$ $(x_0 = [\varphi_0, \varphi_0, \theta_0]T)$, and the equilibrium condition of motion is $f(x_0, u_0) = 0$. For the controlled model $m_j$ $(j = 1, 2,..., m)$ of the bionic robot, $\forall m_j \in M \forall m_j \in M$, the pose prediction probability of binocular vision navigation is

$$\bar{c}_j = \sum^m P_{ij} u_i(k-1) \tag{5}$$

The lateral deflection transition probability of the robot pose is:

$$u_{i/j}(k-1/k-1) = P\left(m_i(k-1)/m_j(k), z^{k-1}\right) \tag{6}$$

Information fusion is performed on the pose parameters of the controlled object, and the mixed input of the parameters is obtained as:

$$\hat{x}^{0j}(k-1/k-1) = \sum_i^m \hat{x}^i(k-1/k-1) u_{i/j}(k-1/k-1) \tag{7}$$

For the binocular visual navigation and tracking model $m_j$ $(j = 1, 2,..., m)$, $\forall m_j \in M$, $\hat{x}^i(0)(k-1/k-1)$ and $p^{0j}(k-1/k-1)$, as the image pixel inputs for visual navigation, are substituted into the disturbance suppression model of the robot, and the filter (usually a Kalman filter) is selected for steady-state tracking fusion filtering. Then, the state estimate $\hat{x}^j(k/k)$ and its estimated covariance are obtained, $p^j(k/k)$. In the control constraint parameter model constructed above, the optimization solution of the control objective function and the control law design of the bionic robot are conducted.

### 3.2. Hierarchical Subdimensional Space Motion Planning

Taking the 6-degree-of-freedom parameters of the robot's rotating joints as control constraints, a hierarchical subdimensional space motion planning model of the robot is established, and a hierarchical subdimensional space motion planning model of the bionic robot in longitudinal motion, lateral motion, and grasping action is constructed. which are described below.

Portrait:

$$\begin{cases} mV\dot{\theta}\cos(\sigma) = F_y \\ J_z\dot{\omega}_{z1} + (J_y - J_x)\omega_{x1}\omega_{y1} = M_{z1} \\ \varphi = \theta + \alpha \end{cases} \tag{8}$$

Lateral:

$$\begin{cases} -mV\sigma = F_z \\ J_y\omega_{y1} + (J_x - J_z)\omega_{z1}\omega_{x1} = M_{y1} \\ \phi = \sigma + \beta \end{cases} \tag{9}$$

Grab:

$$J_x\dot{\omega}_{x1} + (J_z - J_y)\omega_{y1}\omega_{z1} = M_{x1} \tag{10}$$

In the space of 6 degrees of freedom, the relationship between each joint circle space of the robot is obtained. Through the inversion control of the actual geometric parameters of the robot, the center side shift and lateral offset are reduced, and the robot pose model $m_j$ at time $k$ is calculated ($j = 1, 2, \ldots, m, m_j \in M$); the likelihood function is:

$$\Lambda_j(k) = P\left(z(k)/m_j(k), z^{k-1}\right) \tag{11}$$

Among these values, $\Lambda_j(k)$ obeys a normal distribution with a mean of 0 and variance of $S^j(k)$, and $S^j(k)$ is the covariance matrix of the positioning error of each joint circle. The end pose correction is performed according to the geometric parameters of the robot, and the correction probability is:

$$u_j(k) = P\left(m_j(k)/z^k\right) \tag{12}$$

The single-joint coordinate data of the robot are tracked in parallel in the robot base coordinate system, and the robot's pose parameters are fed back to the hierarchical subdimensional space motion planning model for error correction.

### 3.3. Robust Control Law

The initial target position distribution coordinate points of the bionic robot are established using the position and posture measurement data output by the binocular vision system. The influence of uncertain disturbance factors of pose is introduced into the hierarchical subdimensional space motion planning model, and the optimal guidance law of robot under binocular vision navigation is obtained as follows:

$$\hat{x}(k/k) = \sum_{j}^{m} \hat{x}^i(k/k)u_j(k) \tag{13}$$

We define the initial state vector $x(t)$ for robot pose determination $= [x(t), y(t), z(t), \dot{x}(t)\dot{y}(t), \dot{z}(t), \ddot{x}(t), \ddot{y}(t), \ddot{z}(t)]^T$. Through the control law of this paper, the pose adaptive correction is conducted, and the control state equation and observation equation are obtained as:

$$x(k+1) = \Phi(k)x(k) + w(k) \tag{14}$$

$$z(k) = H(k)x(k) + v(k) \tag{15}$$

Among these values, $w(k)$ is the fusion Kalman weighting coefficient, the covariance matrix is $Q(k)$, $v(k)$ is the optimal pose angle output vector after fusion filtering, and the motion attribute covariance matrix of the output target is $R(k)$.

Through binocular vision navigation control, the improved extended Kalman filter (EKF) is adopted to realize the adaptive correction of the position and posture of the bionic robot and to improve the stability and accuracy of robot positioning and tracking.

## 4. Numerical Results and Analysis

### 4.1. Simulation Experiment

To test the performance of the control algorithm designed in this paper in realizing the robust control of the bionic robot and the correction of the pose and orientation, the simulation experiment was analyzed, and the simulation platform for the experiment was Matlab, version 7. The measurement accuracy range of the binocular vision navigation system was 0.025 mm, the dynamic tracking frequency of the robot pose parameters was 30 Hz, the measurement range of the digital accelerometer was $\pm2/\pm4/\pm8$ g, and the sensitivity of the digital magnetometer was 8 mGauss 12. The sampling period of the robot pose was 0.02 s, the Kalman filtering period was 0.25 s, and the initial values of the pitch angle and yaw angle of the robot action were $\Delta1 = 5°$ and $\Delta2 = 8°$, respectively.

According to the above simulation environment and parameter settings, the simulation analysis of robot pose control was performed. First, the optical CCD binocular vision dynamic tracking system was used to measure the terminal pose of the bionic robot, and the sampling data were obtained, as shown in Figure 6.

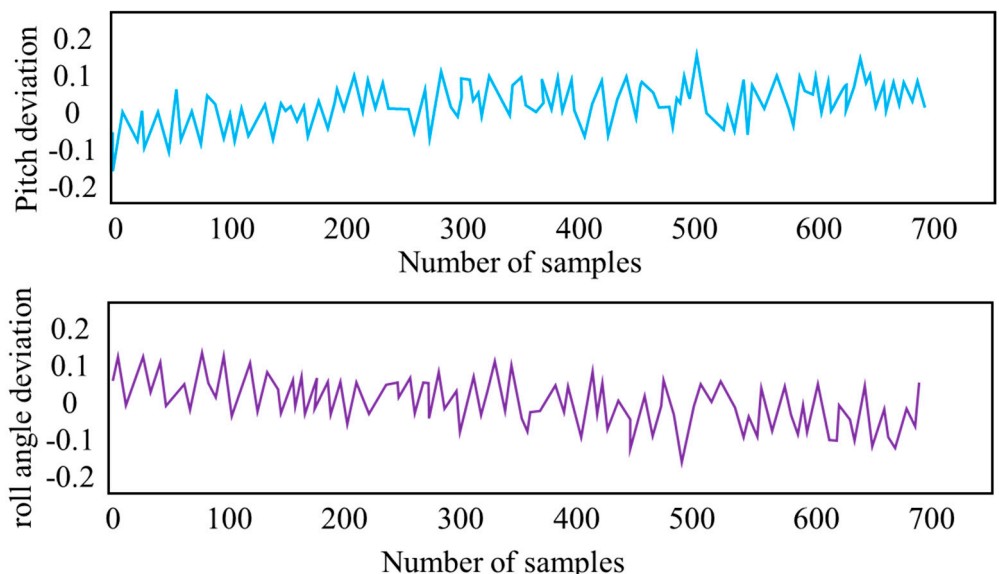

**Figure 6.** The end pose measurement of the bionic robot.

Taking the end position and pose parameter data of the bionic robot shown in Figure 6 as the research model, we address the pose determination control of the bionic robot and obtain the geometric parameters of the pose tracking control of each measuring point, as shown in Table 2.

**Table 2.** Geometric parameters of robot pose determination and tracking.

| Times | Left Arm/mm | Right Arm/mm | Left Leg/mm | Right Leg/mm |
|---|---|---|---|---|
| 1 | 136.650 | 276.320 | 148.543 | 450.687 |
| 2 | 136.743 | 276.335 | 148.665 | 450.688 |
| 3 | 160.021 | 276.236 | 148.578 | 450.676 |

**Table 2.** *Cont.*

| Times | Left Arm/mm | Right Arm/mm | Left Leg/mm | Right Leg/mm |
|:---:|:---:|:---:|:---:|:---:|
| 4 | 160.023 | 276.285 | 148.568 | 450.376 |
| 5 | 136.945 | 276.516 | 148.644 | 450.678 |
| 6 | 136.767 | 276.713 | 148.231 | 450.690 |
| 7 | 136.726 | 276.520 | 148.226 | 450.988 |
| 8 | 136.765 | 276.628 | 148.043 | 450.987 |
| 9 | 160.365 | 276.403 | 148.665 | 450.586 |
| 10 | 160.083 | 276.189 | 148.179 | 450.797 |
| Mean value | 136.967 | 276.408 | 148.765 | 450.664 |

According to the results in Table 2, the average error of the positioning accuracy of the left arm pose transition is 3.453 mm, and the standard deviation is 0.132 when the robot pose positioning and tracking are conducted using the method in this paper. The average error of the positioning accuracy of the right arm pose transition is 2.332 mm, and the standard deviation is 0.145. The average error of the positioning accuracy of the left leg pose transition is 2.432 mm, and the standard deviation is 0.212. The average error of the positioning accuracy of the right leg pose transition is 1.543 mm, and the standard deviation is 0.132. The average positioning error of the robot's pose is increased 2.5 times, and the average error is reduced by 12% when the error correction method is adopted in this paper.

Different methods to accurately measure the end pose of bionic robots include mechanical measurement, visual measurement, and inertial measurement [16]. Mechanical measurement has high accuracy and stability, but its adaptability to environments and pose changes is poor. Visual measurement has high flexibility and adaptability, but it has great influence on factors such as illumination and occlusion. Inertial measurement has fast response speed and good adaptability, but there are problems, such as sensor drift and error accumulation. Therefore, when selecting measurement methods, comprehensive consideration and selection should be applied according to specific application scenarios and requirements. To compare the algorithm's performance, the improved EKF fusion filter and the traditional accelerometer [17] and magnetometer direct calculation method [18] are used to compare the control accuracy of the robot's various attitude angles. The results are shown in Figure 7. It can be seen from the analysis that the fitting error of the robot's end position and attitude parameters is low, the dynamic tracking performance is good, and the robustness is high when using this method to control the bionic robot.

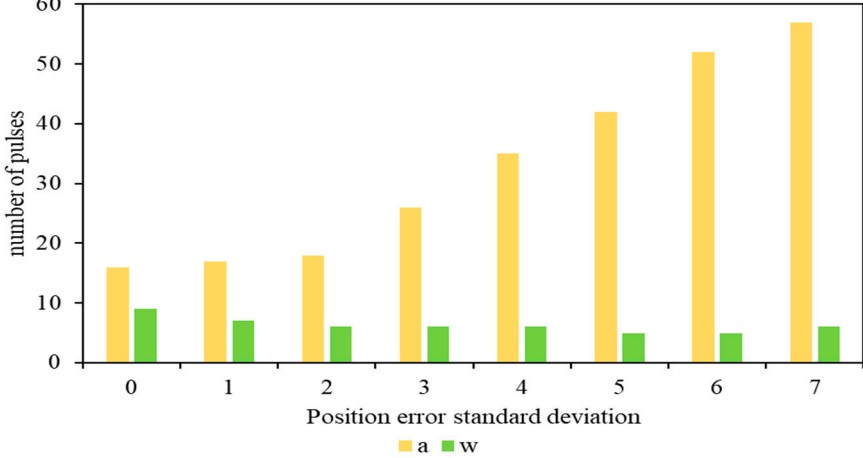

**Figure 7.** Comparison of control accuracy.

Analysis of Joint Driving Torque

The joint drive of the bionic mechanical dog is ultimately driven by the driving torque at the joint, which can be analyzed according to the torque function of each joint [19]. We use the red, dotted line to represent the torque curve of the hip joint and the change in the knee joint. The curve can be viewed as a solid black line for analysis, as shown in Figure 8.

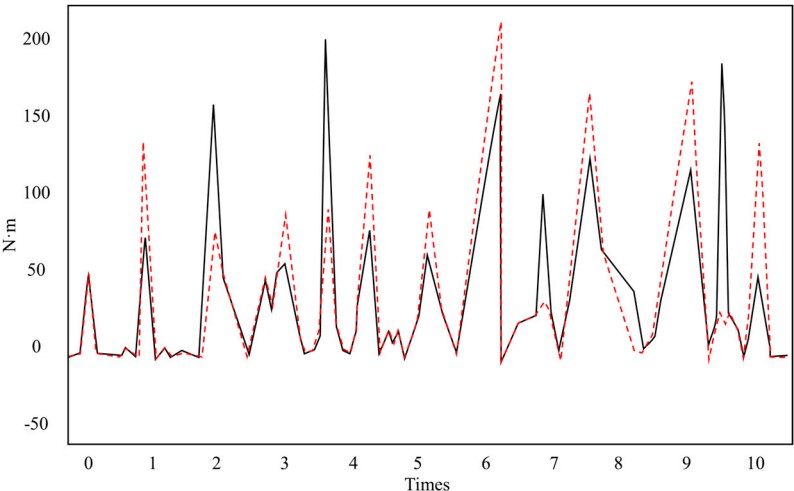

**Figure 8.** Driving moment curves of the hip and knee joints of the right hind leg.

From the above figures, it can be seen that the driving torque changes suddenly during the walking of the bionic robot dog. The smoother the driving torque curve of the joint, the more stable the motion of the robot [20]. It can be seen from the peak of the driving torque curve of the hip joint and knee joint of the right rear leg that the bionic robot dog has a torque imbalance phenomenon in the process of walking. In addition, when a certain phase is in the ground phase, its torque is much greater than when it is in the swing phase. Therefore, the mutation and vibration of the driving torque can be reduced by optimizing the driver control algorithm to make the driving torque curve smoother, thus improving the stability of the robot.

The factors that affect the response time of the angle θ and two-dimensional trajectory mainly include the inertia and friction of the mechanical structure, the response speed and accuracy of the sensor, and the complexity and accuracy of the control algorithm [21]. For example, the influence of sensors and control algorithms on the control accuracy and response time of the robot system can be analyzed by establishing schematic diagrams of sensors and control algorithms. Through the analysis and modeling of the schematic diagram, we can better understand the interaction between different factors in the robot system so as to realize optimization and control of the system. The curve simulation diagram is shown in Figure 9.

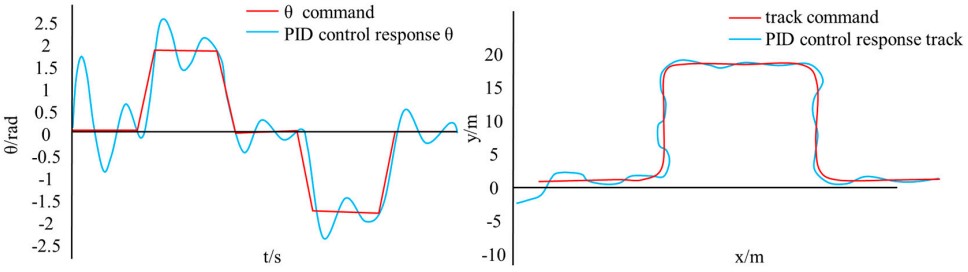

**Figure 9.** Schematic diagram of the angle θ response time and two-dimensional trajectory tracking of the PID algorithm when the broken line moves.

Through analysis of the *x*-axis and the *y*-axis, the PID algorithm has a certain following effect in the movement of the linear *x*-axis. It starts to stabilize after about 15 s on the *y*-axis, the $\theta$ angle also tends to be stable after 20 s, and the response time is relatively short. The error simulation is shown in Figure 10.

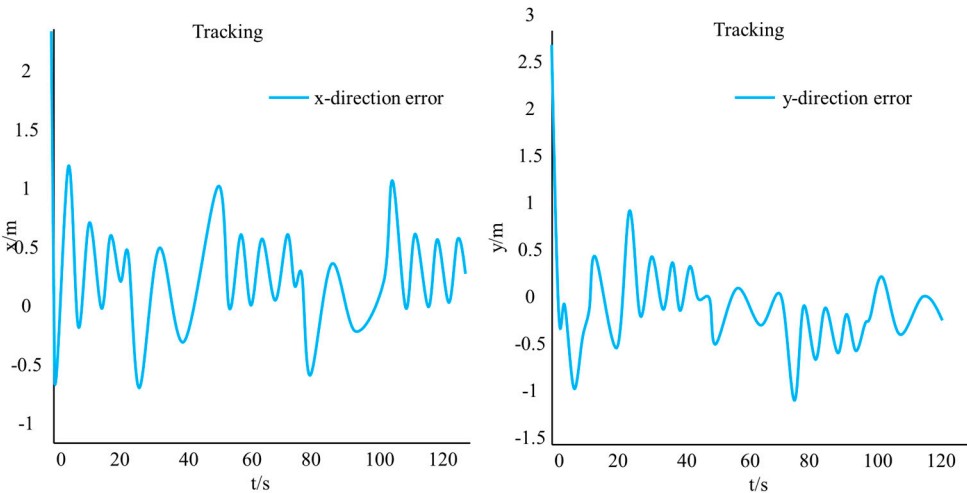

**Figure 10.** *x*, *y*, and $\theta$ axis trajectory tracking error.

It can be seen from the broken line error simulation that the anti-interference of the PID controller is not very strong, and the response time is slow. The simulation range of the $\theta$ axis is $(-0.615$–$0.875)$ m. Therefore, considering the error and control precision, we need to find a better algorithm to improve the control precision and reduce the error.

## 5. Conclusions

This paper studies the pose positioning and optimal control of bionic robots and proposes a robust control algorithm for bionic robots based on binocular vision navigation. An optical CCD binocular vision dynamic tracking system is used to measure the terminal pose parameters of the bionic robot, establish the kinematic model and control constraint parameter model of the controlled object, and perform pose adaptation in the robot's hierarchical subdimensional space motion planning model. The modified EKF fusion filter is used to optimize the control parameters, realizing the self-adaptive correction and robust control of the bionic robot. The experimental analysis shows that the methods in this paper have low fitting error to the robot's end pose parameters, better dynamic tracking performance, and high control robustness.

In the future, on the basis of continuously optimizing the gait planning algorithm, we will gradually add diagonal trot gait and even jumping gait. If conditions permit, sensors, such as depth cameras and laser radar, can be added, and the leg joint structure of the bionic mechanical dog needs to be continuously optimized with reference to dogs in real life.

**Author Contributions:** Conceptualization, Z.L. and X.P.; Methodology, N.X., X.Z. and Y.S.; Formal analysis, Z.L., N.X., X.P. and Y.S.; Investigation, X.Z.; Data curation, N.X.; Writing — original draft, Z.L., X.Z., X.P. and Y.S. All authors have read and agreed to the published version of the manuscript.

**Funding:** The authors would like to acknowledge financial support from the Chinese Aeronautical Establishment "Research on Inertial-Based Vision Aided Autonomous Navigation System (Aeronautical Science Foundation of China)" (Grant No: 201958068002).

**Institutional Review Board Statement:** Not applicable.

**Data Availability Statement:** The figures used to support the findings of this study are included in the article.

**Acknowledgments:** The authors would like to express sincere thanks to those technicians who contributed to this research.

**Conflicts of Interest:** The authors declare that they have no conflict of interest.

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
