# Peer review of "Motion Control Method of Bionic Robot Dog Based on Vision and Navigation Information"

_applsci, doi:10.3390/app13063664_

Round 1

Reviewer 1 Report

“Motion control method of bionic robot dog based on vision and navigation information” is the title of  the authors work. The concept is strong, and the presentation is nicely organized. To make the work more accessible, I suggest putting in additional effort in relation to the following aspects:

1. The introduction is weak, and the authors are jumping from one part to the other part very rapidly without a deep investigation of references.

2. There is no clear information on how this work is novel from other works. The contribution should be highlighted (in the form of points) at the end of the section 1.

3. Since the topic of the article is well-researched, the cited references are not enough in this regard. So, the authors are invited to add more references to cover many related approaches over the past years (10.1155/2018/2572986)

4. what is  ANSYS ?

5. there are many notations that are badly presented for example in lines 209, 213, 214, 222, 237, 238, 239, etc.. Authors should check the notation over the hole paper.

6. How can the motion control method of a bionic robot dog be optimized by combining vision and navigation information?

7. What are the different methods for accurately measuring the end pose of a bionic robot and how do these methods compare in terms of accuracy and reliability?

8. How can the analysis of joint driving torque in a robotic system be used to improve the design and control of the system for better performance and energy efficiency?

9. How do the driving moment curves of the hip and knee joints of the right hind leg of a quadrupedal robot affect its locomotion and stability, and how can they be optimized for better performance in various tasks and environments?

10. What factors affect the response time of the angle θ and the two-dimensional trajectory in a robotic system, and how can a schematic diagram be used to model and analyze these factors for better performance and control of the system?

11. Results and discussion section must be improved by adding a discussion and comparing results with previous studies published in the field.

12. Some future directions should be added to the conclusion section.

Author Response

Reply

  1. The introduction has adjusted and supplemented the research content.
  2. To solve these problems, this paper proposes a robust control algorithm for bionic robot based on binocular vision navigation. Firstly, the binocular vision dynamic tracking system is used to collect the pose parameters and build the kinematics model; then, using binocular vision tracking method to realize the adaptive correction of the position and posture of the bionic robot and the improved design of the control law; finally, simulation experiments are carried out to demonstrate the superior performance of this method in improving the navigation and tracking accuracy of the robot and reducing the end position and attitude control error.
  3. Thank you for your suggestion. References have been added.
  4. ANSYS is a finite element analysis software.
  5. The notations in the article have been checked and modified.
  6. The initial target position distribution coordinate points of the bionic robot are established by using the position and posture measurement data output by the binocular vision system. The influence of uncertain disturbance factors of position and posture is introduced into the hierarchical sub-dimensional space motion planning model, and the optimal guidance law of the robot is obtained under binocular vision navigation. Through binocular vision navigation control, the improved Extended Kalman Filter (EKF) is adopted to realize the adaptive correction of the position and posture of the bionic robot and improve the stability and accuracy of robot positioning and tracking.
  7. Different methods to accurately measure the end pose of bionic robot include mechanical measurement, visual measurement and inertial measurement. Mechanical measurement has high accuracy and stability, but its adaptability to environment and attitude changes is poor. Visual measurement has high flexibility and adaptability, but it has great influence on factors such as illumination and occlusion. Inertial measurement has fast response speed and good adaptability, but there are problems such as sensor drift and error accumulation. Therefore, when selecting measurement methods, comprehensive consideration and selection should be made according to specific application scenarios and requirements.
  8. According to the distribution and size of joint driving torque, the optimization direction of mechanical structure can be determined, for example, by reducing mechanical friction, optimizing transmission ratio and other ways to reduce joint driving torque. By analyzing the variation trend and amplitude of the joint driving torque, the optimization direction of the control strategy can be determined, for example, by dynamically controlling the joint driving torque to reduce energy consumption and vibration.
  9. The smoother the driving torque curve of the joint, the more stable the motion of the robot. It can be seen from the peak of the driving torque curve of the hip joint and knee joint of the right rear leg, which means that the bionic robot dog has a torque imbalance phenomenon in the process of walking. In addition, when a certain phase is in the ground phase, its torque is much greater than when it is in the swing phase. Therefore, the mutation and vibration of the driving torque can be reduced by optimizing the driver control algorithm to make the driving torque curve smoother, thus improving the stability of the robot.
  10. The factors that affect the response time of the angle θ and two-dimensional trajectory mainly include the inertia and friction of the mechanical structure, the response speed and accuracy of the sensor, and the complexity and accuracy of the control algorithm. For example, the influence of sensors and control algorithms on the control accuracy and response time of the robot system can be analyzed by establishing schematic diagrams of sensors and control algorithms. Through the analysis and modeling of the schematic diagram, we can better understand the interaction between different factors in the robot system, so as to realize the optimization and control of the system.
  11. In order to compare the algorithm performance, the improved EKF fusion filter and the traditional accelerometer and magnetometer direct calculation method are used to compare the control accuracy of the robot's various attitude angles. The results are shown in Figure 4. It can be seen from the analysis that the fitting error of the robot's end position and attitude parameters is low, the dynamic tracking performance is good, and the robustness is high when using this method to control the bionic robot.
  12. For the future, on the basis of continuously optimizing the gait planning algorithm, we will gradually add diagonal trot gait, even jumping gait, etc. If conditions permit, sensors such as depth camera and laser radar can be added, and the leg joint structure of the bionic mechanical dog needs to be continuously optimized with reference to the dog in life.

Reviewer 2 Report

Motion control method of bionic robot dog based on vision and navigation information

This paper is really poor in many aspects such as presentation, context and evaluation. From this point of view, my decision about the paper is to reject it.

Abstract is not showing the key points. It is really poor and doesnt include main findings.

Introduction was written by randomly, without tracking any methodology, there is no real effort, this was written in a hurry.

The introduction is finished instantly..

This is a real evidence that there is no good explanations for this subject, if one cannot explain the points in introduction section, then it is nearly impossible to make it a good paper.

There is no figure and photo from experimantal stage. How is it possible? How can we know that the authors really make the experiments? The thing i mentioned in introduction still continues in this section.

The references should be reviewed again.

This paper have many doubts. Almost no citations were made in results. How the authors reach these results and comment on it.

If a journal like Applied sciences will accept this paper, it should be deleted thoroughly and written again.

Author Response

Motion control method of bionic robot dog based on vision and navigation information

This paper is really poor in many aspects such as presentation, context and evaluation. From this point of view, my decision about the paper is to reject it.

Abstract is not showing the key points. It is really poor and doesnt include main findings.

Reply: With the progress and development of AI technology and industrial automation technology, AI robot dogs are widely used in engineering practice to replace human beings in high-precision and tedious industrial operations. The bionic robot is easy to produce control error due to the influence of spatial disturbance factors in the process of pose determination. It is necessary to calibrate the robot accurately to improve the positioning control accuracy of the bionic robot. Therefore, a robust control algorithm for the bionic robot based on binocular vision navigation is proposed. The optical CCD binocular vision dynamic tracking system is used to measure the end position and pose parameters of the bionic robot, and the kinematics model of the controlled object is established. Taking the degree of freedom parameter of the robot's rotating joint as the control constraint parameter, the hierarchical sub-dimensional space motion planning model of the robot is established. The binocular vision tracking method is used to realize the adaptive correction of the position and posture of the bionic robot and achieve the robust control. The simulation results show that the fitting error of the robot's end position and attitude parameters is low and the dynamic tracking performance is good when the method is used for the position positioning of the bionic robot control.

Introduction was written by randomly, without tracking any methodology, there is no real effort, this was written in a hurry. The introduction is finished instantly. This is a real evidence that there is no good explanations for this subject, if one cannot explain the points in introduction section, then it is nearly impossible to make it a good paper.

Reply: The introduction has been supplemented.

There is no figure and photo from experimantal stage. How is it possible? How can we know that the authors really make the experiments? The thing i mentioned in introduction still continues in this section.

Reply: Tables and figures in the experimental stage have been added.

The references should be reviewed again.

Reply: References have been updated.

This paper have many doubts. Almost no citations were made in results. How the authors reach these results and comment on it.

Reply: References have been added to the results.

If a journal like Applied sciences will accept this paper, it should be deleted thoroughly and written again.

Reply: The article has been readjusted and supplemented.

Reviewer 3 Report

Comments:

1)

Row 37 dog, and improve the positioning and tracking of the robot dog's end pose [5]. ability. It is

To be

Row 37 dog, and improve the positioning and tracking of the robot dog's end pose ability [5]. It is

2) At the end of the section: introduction, will be good to have concluding observations regarding the used references.

3) Can use legend into figure 1 to explain FL, FR, HL, HR

4) The section: ANSYS finite element analysis

This is too general and should be deleted.

165 ANSYS is a finite element analysis software that initially provides structural linear 166 analysis and thermal analysis. Problems in the fields of physics and other fields are now widely 167 used in the analysis fields of nuclear industry, railway, petrochemical industry, aerospace, 168 automobile transportation, machinery manufacturing, etc. ANSYS provides interfaces for 169 many CAD software, so that the advantages of different software can be fully utilized. Targeted 170 use plays a very important role in the early stage of engineering analysis. 171 ANSYS Workbench is a new platform launched by ANSYS in 2002. The platform 172 integrates the models of 3D modeling software such as Solidworks, CATIA, etc. into ANSYS, 173 and its software operation interface is very friendly, easy to operate and understand, and very 174 convenient. Engineers adapt quickly. ANSYS Workbench actually includes a number of 175 analysis processing units, such as the familiar geometric modeling unit, finite element analysis 176 unit processing module, and Design Xplorer module unit. We can convert the Solidworks 177 model into a parasolid file and save it, and then you can Analyzed in ANSYS Workbench.

5) The section: ANSYS finite element analysis, is described to much generally, without figures for stress distribution or deformation  distribution, without 3d Model of dog, without initial and boundary conditions, safety factor, without figure for dynamic mechanical analysis, etc. The section with finite element analysis needs to be refined.

6) This text can be use in the introduction, but not in the section ANSYS finite element analysis.

186 In the process of human beings exercising their subjective initiative, there are many dangerous 187 places that humans cannot safely reach, such as forest fire inspections, planetary exploration, 188 military material transportation, earthquake disaster relief, polluted nuclear field experiments, 189 demining and detonation, and deep-sea exploration, etc. Bionic robots play an irreplaceable 190 role in performing this series of arduous and dangerous tasks. However, at present, the related 191 technologies of bionic mechanical dogs are not mature, and countries are constantly increasing 192 R&D investment in this field. The research on mechanical dogs has great practical significance 193 for improving the country's soft and hard power and human security, and has important 194 practical application value.

7) Where is shown kinematic model (3d CAD model) of the robot in the article? Really, I don't know how looks the robot dog?

197 In order to realize the robust tracking control of the binocular vision navigation of the 198 bionic robot, it is necessary to first construct the kinematic model of the robot

8) The authors can show how looks "CCD binocular vision dynamic tracking system" with picture, can give in the paper some technical data for "CCD binocular vision dynamic tracking system" which is used in the experimental section?

9) In equation 1 are not explain the variables in detail.

10) In the section: 297 Experimental results and analysis.

The title for this section is Experimental results, but below in the text the authors explain about MatLab simulations. Probably the authors think about numerical results?

Author Response

Reply

  • Therefore, it is necessary to optimize the control of the robot dog, reduce the geometric error of the robot dog, and improve the positioning and tracking ability of the robot dog's end posture [5].
  • To solve these problems, this paper proposes a robust control algorithm for bionic robot based on binocular vision navigation. Firstly, the binocular vision dynamic tracking system is used to collect the pose parameters and build the kinematics model; then, using binocular vision tracking method to realize the adaptive correction of the position and posture of the bionic robot and the improved design of the control law; finally, simulation experiments are carried out to demonstrate the superior performance of this method in improving the navigation and tracking accuracy of the robot and reducing the end position and pose control error.
  • A robot in a crawling state will always have at least three legs or four legs, as shown in Figure 1. One leg is lifted, and the other three legs form a stable triangle area (FL, FR, HL) for the support leg, so the crawling gait has high stability and strong robustness. FL, FR, HL and HR represent left foot, right foot, left hand and right hand respectively.
  • The relevant paragraph has been deleted.
  • Relevant models and data have been added to the finite element analysis section.
  • This part has been adjusted.
  • The kinematics model of the robot has been added.
  • Technical data has been provided in the form of Table 2.
  • Among them, wi(k) and vi(k) are the poses of the joints and the disturbance noise that affects the robot's pose determination. Inverse kinematics is used to perform the dynamic tracking of the robot's binocular vision, and the resulting disparity measure information is Qi(k) and Ri(k).
  • Yes, I have changed it to " Numerical results and analysis".

Round 2

Reviewer 1 Report

The work has been greatly improved. 

Reviewer 2 Report

Accepted

Reviewer 3 Report

Excellent!